# Fisher Information as General Metrics of Quantum Synchronization

**DOI:** 10.3390/e25081116

**Published:** 2023-07-26

**Authors:** Yuan Shen, Hong Yi Soh, Leong-Chuan Kwek, Weijun Fan

**Affiliations:** 1School of Electrical and Electronic Engineering, Nanyang Technological University, Block S2.1, 50 Nanyang Avenue, Singapore 639798, Singapore; 2National Institute of Education, Nanyang Technological University, 1 Nanyang Walk, Singapore 637616, Singapore; 3Centre for Quantum Technologies, National University of Singapore, Singapore 117543, Singapore; 4MajuLab, CNRS-UNS-NUS-NTU International Joint Research Unit, UMI 3654, Singapore 117543, Singapore

**Keywords:** quantum synchronization, Fisher information, quantum information theory

## Abstract

Quantum synchronization has emerged as a crucial phenomenon in quantum nonlinear dynamics with potential applications in quantum information processing. Multiple measures for quantifying quantum synchronization exist. However, there is currently no widely agreed metric that is universally adopted. In this paper, we propose using classical and quantum Fisher information (FI) as alternative metrics to detect and measure quantum synchronization. We establish the connection between FI and quantum synchronization, demonstrating that both classical and quantum FI can be deployed as more general indicators of quantum phase synchronization in some regimes where all other existing measures fail to provide reliable results. We show advantages in FI-based measures, especially in 2-to-1 synchronization. Furthermore, we analyze the impact of noise on the synchronization measures, revealing the robustness and susceptibility of each method in the presence of dissipation and decoherence. Our results open up new avenues for understanding and exploiting quantum synchronization.

## 1. Introduction

Synchronization is an emergent dynamic process that explains numerous phenomena such as the flashing of fireflies (*Photinus carolinus*) in tandem [1], the clicking of pacemakers [2], and the unusual sideward swaying of the Millenium bridge in London [3]. A key feature of synchronization is the existence of self-sustaining oscillators coupled to each other or to a driven oscillator.

Synchronization has yielded many interesting mechanisms for complex systems: limit cycles, amplitude death, oscillation death, and so forth. Synchronization is also intimately connected to chaos theory, where one speaks of the synchronization of chaotic oscillators. Typically, synchronization revolves around the study of the behavior of an oscillatory system under a periodic external force or the interaction between two or more coupled oscillatory systems. In this regard, many different oscillators have been studied: Stuart–Landau oscillators [4], Duffing oscillators [5], van der Pol oscillators [6], Kuramoto oscillators [7], Rössler oscillators [8], and so forth. Depending on the situation and context, different measures are proposed for measuring synchronization [9,10].

In recent years, there has been extensive work on quantum versions of classical synchronization [11,12,13,14]. Instead of the classical phase space, one can investigate the Wigner function and probe into the presence of an Arnold-like tongue. For more than one oscillator, measures have been devised to detect the presence of quantum synchronization [15,16]. As in the classical case, we know less about quantum mixed synchronization. In the classification of measures for continuous-variable quantum systems, authors have briefly mentioned mixed synchronization for two oscillators with opposite momenta and its deep connection to Einstein–Podolsky–Rosen pairs [16].

Moreover, in the quantum regime, it is known that a limit-cycle oscillator with a squeezing Hamiltonian can undergo a bifurcation where the Wigner function splits into two symmetrical peaks [17]. In Ref. [18], the authors have investigated two quantum oscillators that display peak synchronization at two different phases, with a phase of π apart. In such cases, the system can be regarded as in-phase synchronization with the drive, but the phase locking happens at two distinct phases. We refer to this phenomenon as 2-to-1 synchronization, and it is similar to the case described as mixed synchronization in Weiss et al. [18].

Fisher information has been used as a measure of the ability to estimate an unknown parameter or as a measure of the state of disorder of a system [19,20,21,22,23]. The quantum Fisher information (QFI) serves as a crucial measure in quantum parameter estimation, providing insights into the precision with which a quantum system can estimate an unknown parameter. QFI finds applications in various areas of quantum information science. In quantum sensing, QFI provides a measure of the sensitivity of a quantum sensor to external perturbations, enabling the development of high-precision measurement devices. QFI is also relevant to many-body entanglement [24,25,26] and quantum error correction [27,28], where accurate estimation of parameters is essential. Despite its wide range of applications, it has never been considered relevant for continuous-variable quantum synchronization.

In this paper, we look at various measures for the quantum synchronization of an externally driven oscillator and explore the possibility of introducing Fisher information to define quantum synchronization, especially for the general case of *n*-to-1 quantum synchronization. This paper is organized as follows: In Section 2, we describe the driven quantum Stuart–Landau oscillator and we discuss various possible measures of synchronization. In Section 3, we study the behavior of the various measures of synchronization and discuss 2-to-1 synchronization in Section 4, the case where a squeezing term is added to the oscillator. There are different possible noises in such systems, and we investigate the effects of noise in Section 5. In Section 6, we compare the correlations between the different measures. Finally, in Section 7, we investigate the asymmetric case of 2-to-1 synchronization and make some concluding remarks in Section 8.

## 2. Oscillator Model and Synchronization Measures

In this study, we investigate the quantum van der Pol oscillator (also known as the quantum Stuart–Landau oscillator [29]) subjected to both single-photon drive and two-photon squeezing drive. Both the van der Pol and Stuart–Landau oscillators are important classes of nonlinear systems classically [2], and both of them have recently been adopted to the quantum domain [12,13,14,30]. The quantum Stuart–Landau oscillator, which is an approximation of the van der Pol oscillator to the first order, was deemed the starting point of continuous-variable quantum synchronization, and still serves as the paradigm in the current literature (but it is often still referred to as the quantum van der Pol oscillator) [11,17,31,32,33]. The master equation in the rotating frame of the drive gives (with ℏ=1)
(1)$$\begin{array}{cc} \dot{\rho }= & -i\left[\hat{H},\rho \right]+\gamma_{1}D\left[a^{†}\right]\rho +\gamma_{2}D\left[a^{2}\right]\rho +\gamma_{3}D\left[a\right]\rho \\ \hat{H}= & \Delta a^{†}a+iE\left(a-a^{†}\right)+i\eta \left(a^{†2}e^{2i\varphi }-a^{2}e^{-2i\varphi }\right), \end{array}$$
where $D\left[L\right]\rho =L\rho L^{†}-\frac{1}{2}\left(L^{†}L\rho +\rho L^{†}L\right)$, γ represents the rate of decay, with $\gamma_{1}$, $\gamma_{2}$, and $\gamma_{3}$ corresponding to negative damping, nonlinear damping, and linear damping, respectively. $\Delta =\omega_{0}-\omega_{d}$ is the amount of detuning between the frequency of the drive, $\omega_{d}$, and the frequency of the oscillator, $\omega_{0}$. *E* is the amplitude of the harmonic drive, with *a* and $a^{†}$ being the annihilation and creation operators. η is the squeezing parameter, with φ representing the phase of squeezing.

In this paper, we focus on the measures of quantum phase synchronization in an externally driven oscillator. As a measure of phase synchronization, the phase coherence is frequently used in the literature and defined as [11,18,33,34]
(2)$$S_{pcoh}=\frac{\mathrm{Tr}[a\rho ]}{\sqrt{\mathrm{Tr}[a^{†}a\rho ]}},$$
where |S| measures the degree of phase coherence with a range of 0≤|S|≤1. Another appropriate simple measure is based on the relative phase distribution [35,36]:(3)$$S_{peak}=2\pi \;\max\left[P\left(\Phi \right)\right]-1,$$
where the phase distribution is defined by P(Φ)=(1/2π)〈Φ|ρ|Φ〉 with $|\Phi \rangle =\sum_{n=0}^{\infty }e^{in\Phi }|n\rangle$. $S_{peak}$ represents the maximum value of P(Φ) compared to a uniform distribution. This measure is valuable for detecting synchronization because it is exclusively nonzero when P(Φ) deviates from a flat distribution.

It is well known that the phase operator is not well defined in quantum theory. However, most quantum harmonic oscillators are populated up to some finite levels, and we can resort to the Pegg–Barnett phase operator. In Ref. [31], the mean resultant length (MRL), which incorporates the Pegg–Barnett operator, has been proposed as a measure of synchronization. It arises from the study of circular statistics [37] and was initially developed for 1-to-1 synchronization. However, it can be generalized to measure n-to-1 synchronization. The *n*-th order mean resultant length ($MRL^{\left(n\right)}$) of a circular distribution is given by
(4)$$MRL^{\left(n\right)}=\sqrt{{\langle \sin n\phi \rangle }^{2}+{\langle \cos n\phi \rangle }^{2}}=\left|\langle e^{in\phi }\rangle \right|.$$

This measure is capable of capturing *n*-to-1 synchronization, which exhibits multiple peaks in the phase distribution P(Φ) and fixed points in the quasi-probability phase-space distribution, e.g., Wigner function.

Fisher information proves to be an important tool for determining classical synchronization in a system of Kuramoto oscillators [38,39]. It has been mooted as a good measure for phase drift in clock synchronization, both classical and quantum [40,41,42,43]. It is also a useful measure in classical signal processing [44], being intimately related to the Cramer–Rao bound. The quantum Fisher information (QFI) is defined as the expectation of $L_{\rho }^{2}$, with $L_{\rho }$ being the Symmetric Logarithmic Derivative [45], which measures the distinguishability in the space of density matrices. For a driven quantum limit-cycle oscillator, in which the probability distribution along the limit cycle is attracted and concentrated onto one (or more) fixed point, measuring such distinguishability is equivalent to measuring the kurtosis [46] of the distribution (how peaked the distribution is). Motivated by these works, we propose the quantum Fisher information as a measure of quantum phase synchronization [47], which is defined mathematically in a compact form:(5)$$QFI=F_{Q}\left[\rho ,\hat{A}\right]=2\sum_{k,l}\frac{{\left(\lambda_{k}-\lambda_{l}\right)}^{2}}{\left(\lambda_{k}+\lambda_{l}\right)}\left|\langle k\right|\hat{A}{\left|l\rangle \right|}^{2},$$
where $\lambda_{k,l}$ and |k,l〉 are the eigenvalues and eigenvectors of the steady state $\rho =\rho_{ss}$. We use $\hat{A}=a^{†}a$ to measure the phase uncertainty in the steady state.

Phase synchronization is closely related to the phase distribution P(Φ), which is a classical probability distribution. Therefore, it make sense to directly inspect this classical distribution to obtain information about synchronization. We propose another measure of phase synchronization using classical Fisher information (CFI). This new measure is defined by the classical Fisher information of the phase distribution P(Φ):(6)$$\begin{array}{c} CFI=E\left[{\left(\frac{\partial }{\partial \Phi }\log P\left(\Phi \right)\right)}^{2}\right], \end{array}$$

It is important to note that this CFI is different from the conventional Fisher information, which is directly calculated from the density matrix as $F\left(\hat{X}|\theta \right)=\sum_{x}\frac{1}{p(x|\theta )}{\left(\frac{\partial p(x|\theta )}{\partial \theta }\right)}^{2}$, where p(x|θ) is the probability of observing outcome *x* when measuring observable $\hat{X}$ [47].

Classical and quantum Fisher information and $S_{peak}$ read 0 for unsynchronized states, but are unbounded for highly synchronized states, whereas phase coherence and $MRL^{\left(n\right)}$ are bounded between 0 and 1.

Two advantages of FI-based measures over phase coherence can be observed: Firstly, Fisher information appears to be more sensitive to highly synchronized states, while exhibiting less sensitivity at the other extreme, as shown in Figure 1. However, in most cases, our primary interest lies in the highly synchronized states. Secondly, FI-based measures are more general metrics of synchronization. Measures such as phase coherence face limitations in detecting the synchronization of squeezed states or, more generally, Wigner functions with multiple peaks— *n*-to-1 synchronization, see Figure 2 as an example. In contrast, FI-based measures are capable of detecting synchronization in such instances. As a measure of synchronization, we find that FI-based measures are not only comparable to existing measures for normal cases of 1-to-1 synchronization; they are also more appropriate for the measurement of 2-to-1 synchronization. Measuring QFI in experiments can be challenging due to its reliance on the full quantum state of the system. However, there are various strategies that have been developed to estimate QFI experimentally, such as randomized measurement [48,49,50].

Recently, there has been some work [51] carried out to relate quantum synchronization to the quantum geometric phase [52]. In this work, they showed that the geometric phase for the quantum Stuart–Landau oscillator under a driven pump exhibits an Arnold-tongue-like structure, somewhat similar to the Arnold tongue in quantum synchronization as measured by the shifted phase distribution of the *Q* function. Also, for two oscillators, it is sometimes useful to measure the quantum mutual information [15,53,54].

## 3. 1-to-1 Synchronization

We first study the scenario of a coherently driven oscillator without a squeezing drive (by simply setting η=0). When only coherent driving is present, there will be only one preferred phase (namely ’fixed point’) to synchronize to and the phase distribution P(Φ) has only one peak, as shown by the first row in Figure 3, whose position indicates the relative phase between the oscillator and drive. With increasing amplitude of the driving, the quantum phase synchronization between the oscillator and drive improves, and so do the values of the synchronization measures. This indicates a monotonic behavior in the measure. We show that all measures qualitatively agree in Figure 3, where synchronization measures are plotted against the coherent driving amplitude *E*. Therefore, these are all valid measures to capture 1-to-1 quantum phase synchronization, and their correlations are close to unity, as shown in the later section. Take note that in Figure 3 the unbounded and bounded measures are plotted separately.

In Figure 3, the synchronization measures are compared across different nonlinear damping ratios $\gamma_{2}/\gamma_{1}$, where this ratio directly controls the radius of the limit cycle and mean photon number in the oscillator. Conventionally, the oscillator is regarded in ’semi-classical’ regime when $\gamma_{2}/\gamma_{1}\approx 1$, and ’quantum’ regime when $\gamma_{2}/\gamma_{1}\gg 1$. We can see that these measures remain valid for different regimes. A driven oscillator with a smaller radius (i.e., larger $\gamma_{2}/\gamma_{1}$) is more prone to lose synchronization by phase diffusion and quantum noise [11,12,13]. Comparing two columns of Figure 3, the values of a synchronization measure are higher in the classical regime, as expected. Note that in the right column of Figure 3, the value of CFI surpasses QFI at certain driving amplitudes *E*. As we have explained previously, the CFI we propose in this paper is not the direct classical analog of QFI. Therefore, this is not a violation of the assertion that QFI should be the supremum of the CFI over all observables.

More insights can be developed in the deep quantum regime ($\gamma_{2}\to \infty$), where the analytical solutions to all these measures can be obtained. By using the 3×3 density matrix ansatz proposed in [31], the analytical equations for $MRL^{\left(1\right)}$, QFI, and phase coherence $S_{pcoh}$ are obtained as follows (with $\Delta =0,\gamma_{1}=1,\gamma_{3}=0$):(7)$$\lim_{\gamma_{2}\to \infty }MRL^{\left(1\right)}=\frac{2E}{9+8E^{2}},$$
(8)$$\lim_{\gamma_{2}\to \infty }QFI=4|\frac{2E}{9+8E^{2}}{|}^{2},$$
(9)$$\lim_{\gamma_{2}\to \infty }S_{pcoh}=\frac{2E}{\sqrt{\left(8E^{2}+9\right)\left(4E^{2}+3\right)}}.$$

Subsequently, the phase distribution P(Φ) can be obtained:(10)$$\lim_{\gamma_{2}\to \infty }P\left(\Phi \right)=\frac{1}{2\pi }\left[1-\frac{4E}{9+8E^{2}}\cos\left(\Phi \right)\right]$$

After deriving the phase distribution P(Φ), the peak of phase distribution $S_{peak}$ and CFI can be easily obtained as
(11)$$\lim_{\gamma_{2}\to \infty }S_{peak}=\frac{4E}{9+8E^{2}},$$
(12)$$\lim_{\gamma_{2}\to \infty }CFI=4\frac{A_{0}+A_{1}E^{2}+A_{2}E^{4}+A_{3}E^{6}+A_{4}E^{8}}{\lambda \left(9+8E^{2}\right){\left(\lambda -9-8E^{2}\right)}^{2}},$$
where
(13)$$\begin{array}{cc} A_{0} & =729(\lambda -9),\\ A_{1} & =108(17\lambda -201),\\ A_{2} & =544(3\lambda -52),\\ A_{3} & =256(2\lambda -67),\\ A_{4} & =-4096,\\ \lambda & =\sqrt{\left(9+4E+8E^{2}\right)\left(9-4E+8E^{2}\right)}. \end{array}$$

These solutions are valid when the driving amplitude E<<1 (see Appendix A), beyond which the density matrix ansatz breaks down.

## 4. Squeezing Enhances 2-to-1 Synchronization

In this section, we show that squeezing drive can create and enhance quantum 2-to-1 synchronization, i.e., synchronization with two distinct fixed points in phase space, and the phase distribution P(Φ) has two distinct peaks. However, as mentioned previously, some synchronization measures are not suitable for measuring this type of synchronization.

As shown in the left column of Figure 4, increasing squeezing sharpens the two peaks in the phase distribution and thus improves mixed synchronization. Here, we need to consider the following question: does E=0, i.e., no drive, make sense for synchronization? We can always regard the squeezing term as a drive. When squeezing is present without a coherent drive, the synchronization can be regarded as between the oscillator and the squeezing drive. This is also considered in Ref. [17] for frequency entrainment (the frequencies of the oscillator and external drive converge). Note that phase coherence $S_{pcoh}$ and $MRL^{\left(1\right)}$ are zero when only squeezing is present, which is expected, as these measures reflect the first off-diagonal elements in the density matrix. On the other hand, $S_{peak}$ and $MRL^{\left(2\right)}$ scale almost linearly with squeezing.

Furthermore, 2-to-1 synchronization can be created out of 1-to-1 synchronization. This is shown in the right column of Figure 4, where in addition to squeezing, a coherent drive with amplitude E=0.5 is present. This coherent drive creates a single peak when squeezing is off or small. When the squeezing is turned up, the single peak splits into two under pitchfork bifurcation, and so do the corresponding Wigner functions [17]. In this scenario, the two measures (phase coherence $S_{pcoh}$ and $MRL^{\left(1\right)}$) which are only capable of measuring 1-to-1 synchronization decrease and appear to change almost linearly with increasing the squeezing parameter. $MRL^{\left(2\right)}$ is a measure dedicated to 2-to-1 synchronization; therefore, it is unsurprising that it only provides partial information when a single peak is present. This explains why $MRL^{\left(2\right)}$ drops to zero at small η and increases linearly afterwards. The measurement of $S_{peak}$ lacks the ability to differentiate between two types of synchronization. Consequently, only the classical and quantum Fisher information measures exhibit a monotonic relationship with respect to squeezing.

## 5. Effects of Noise

In this section, we investigate and compare the effect of different noises across these measures. We consider two types of noise, namely single photon dissipation and white noise.

The single-photon dissipation process is implemented by the Lindblad dissipator proportional to $\gamma_{3}$ in the master Equation (Equation 1). In Figure 5, all six measures are captured in the surface plots with respect to the single-photon dissipation $\gamma_{3}$ and coherent driving amplitude *E*. It is known that single-photon dissipation can be beneficial for 1-to-1 synchronization in coherently driven oscillators [31,33], which is reflected in Figure 5 among all measures consistently. Surprisingly, this noise-induced synchronization boost is absent in 2-to-1 synchronization, as shown in Figure 6, in which the squeezing η is increasing instead of the driving amplitude *E*. Again, the phase coherence and $MRL^{\left(1\right)}$ remain 0 for the same reason explained in the previous section.

To introduce white noise into the density matrix, we define a noise parameter p∈[0,1]; thus, the noisy steady-state density matrix is defined as
(14)$$\rho_{noisy}=\left(1-p\right)\rho_{ss}+p\hat{I}/N_{dim},$$
where $\rho_{ss}$ is the noiseless steady-state density matrix and $\hat{I}$ is the identity matrix with dimension $N_{dim}$.

After introducing white noise, it is expected for all measures to degrade with increasing *p* as shown in Figure 7 and Figure 8. Interestingly, phase coherence turns out to be the most sensitive to white noise, as shown in Figure 7, where there is a bigger drop in the measure as a function of noise *p* compared to other measures.

## 6. Correlations between Measures

In this section, a correlation analysis was performed to investigate the extent to which the different measures of quantum synchronization carry independent and nonredundant information. We calculate the Pearson correlation between the values of different measures, defined as
(15)$$C=\frac{\mathrm{cov}(X,Y)}{\sigma_{X}\sigma_{Y}},$$
with cov(X,Y) being the covariance between two synchronization measures and σ the standard deviation. In the case of 1-to-1 synchronization, i.e., a single peak, high Pearson correlations are observed across all the measures, as shown in Figure 9a, whereas in the case of 2-to-1 synchronization, it is obvious that phase coherence $S_{pcoh}$, $S_{peak}$, and $MRL^{\left(1\right)}$ are ill-suited measures, as they are negatively related to the other three proper measures, as shown in Figure 9b.

From both plots, we can tell the connections between these measures: CFI, QFI, and $MRL^{\left(2\right)}$ are highly correlated in their response to the driving. On the other hand, phase coherence $S_{pcoh}$ and $MRL^{\left(1\right)}$ exhibit a strong connection, as they are both related to the first off-diagonal coherences.

## 7. Asymmetrical Synchronization

So far, we have discussed cases when the two peaks in phase distribution are symmetrical (i.e., the peaks have identical amplitudes). To complete the whole picture, in this section we discuss the situation when the two peaks are distorted and asymmetrical. This asymmetrical phase distribution can be observed when both coherent drive and squeezing are present with a difference of phase, as shown in Figure 10.

Interestingly, both FI-based measures are shown to be insensitive to the change in symmetry by varying the phase of squeezing φ in Figure 10. Meanwhile, all the other measures have great dependence on the phase of squeezing φ. This is another convenient trait of FI-based measures, being tolerant to phase mismatch. As the phase of squeezing is usually determined by the specific experiment setup, such as the properties of cavity in the cQED platform [55] and nonlinear crystals in the optical platform [56,57].

## 8. Concluding Remarks

In conclusion, this research provides a comprehensive analysis of quantum phase synchronization measures. Our work proposes a novel approach to measure the degree of synchronization by deploying classical and quantum Fisher information. Significantly, both measures demonstrate success in characterizing both the 1-to-1 and 2-to-1 synchronization regimes, where other existing methods fail to yield reliable results in one or another.

Our comparative study of the classical and quantum Fisher information measures with existing measures highlights the advantages and limitations of each method. Our study offers valuable guidance for future investigations and practical implementations. Our analysis of the impact of noise on the synchronization measures reveals the robustness and susceptibility of each method in the presence of decoherence. Furthermore, the correlations between these measures provide insight into the similarities and differences between different measures of quantum synchronization.

Our findings contribute significantly to the characterization of quantum phase synchronization, particularly in the 2-to-1 synchronization regime. These results pave the way for further research in the field, such as the development of more efficient and robust quantum communication and computing protocols. Future work could explore other synchronization regimes, investigate the impact of various types of noise, and assess potential applications of our proposed measures in real-world quantum systems.

## Figures and Tables

**Figure 1 entropy-25-01116-f001:**
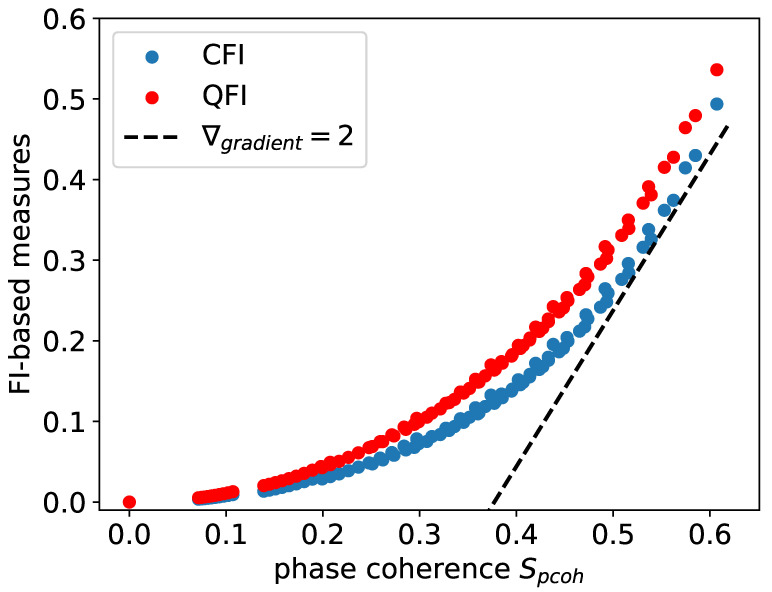
Phase coherence vs. FI-based measures. It is interesting to see FI-based measures are more sensitive (larger gradient) for highly synchronized states, as shown by the dotted reference line indicating $\nabla_{gradient}=2$. Sample data simulated with $\Delta =0,\;E=0.5,\;\eta =0,\;\varphi =\pi /2,\;\gamma_{1}=1,\;\gamma_{2}\in \left[1,10\right],\;\gamma_{3}=0$.

**Figure 2 entropy-25-01116-f002:**
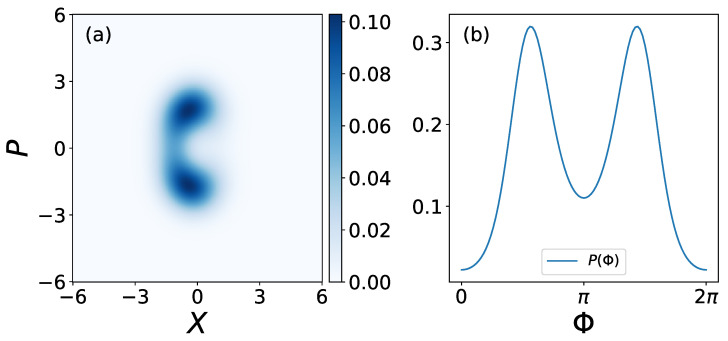
An example of a squeezed steady-state (**a**) Wigner function and its corresponding (**b**) phase distribution P(Φ). Squeezed Wigner function and phase distribution have two distinct peaks, which we refer to as 2-to-1 synchronization. Parameters in this example: Δ=0, E=η=0.5,
φ=π/2,
$\gamma_{1}=\gamma_{2}=1,\;\gamma_{3}=0$.

**Figure 3 entropy-25-01116-f003:**
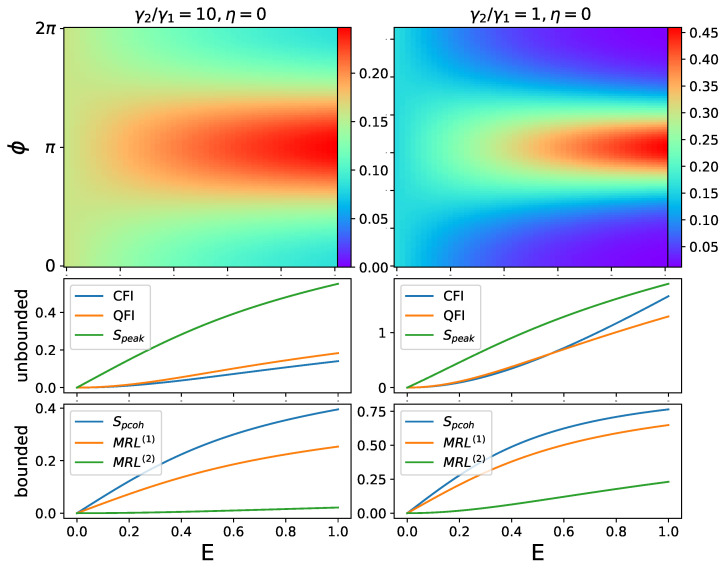
Phase distribution P(ϕ) and synchronization measures plotted against driving amplitude *E*. Fixed parameters: $\Delta =0,\;\gamma_{1}=1,\;\gamma_{3}=0$. In these cases of 1-to-1 synchronization, the driven oscillator has only one preferred phase to synchronize. Unbounded and bounded measures are plotted separately.

**Figure 4 entropy-25-01116-f004:**
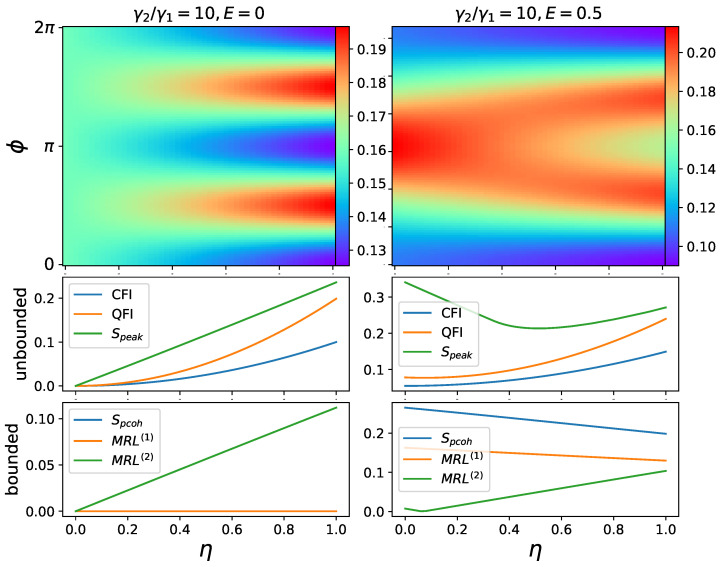
Phase distribution P(ϕ) and synchronization measures plotted against squeezing amplitude η. Fixed parameters: $\Delta =0,\;\varphi =\pi /2,\;\gamma_{1}=1,\;\gamma_{3}=0$. In these cases of 2-to-1 synchronization, the driven oscillator has two distinct phases to synchronize. Unbounded and bounded measures are plotted separately.

**Figure 5 entropy-25-01116-f005:**
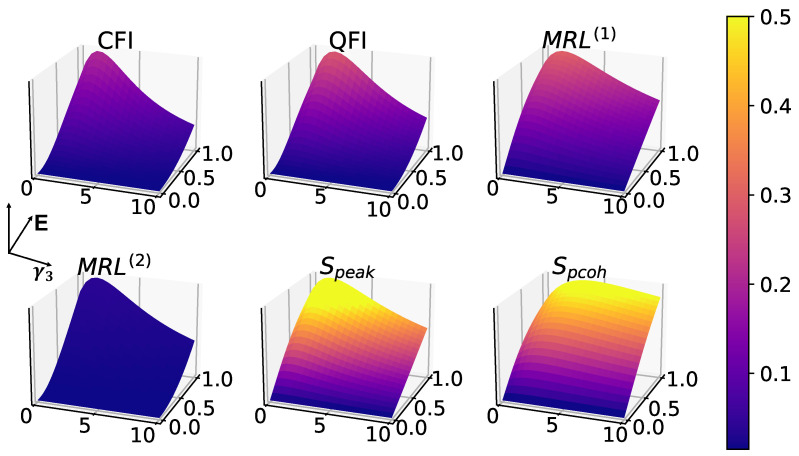
Effects of single-photon dissipation noise in 1-to-1 synchronization, in the absence of squeezing (η=0), with fixed parameters: $\Delta =0,\;\gamma_{1}=1,\;\gamma_{2}=10,\;p=0$. All measures exhibit a noise-induced boost where the dissipation is small, which is consistent with previous work [31,33].

**Figure 6 entropy-25-01116-f006:**
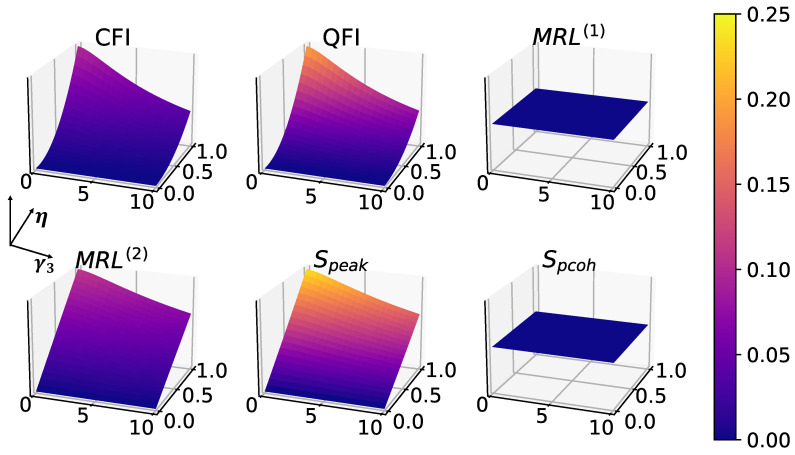
Effects of single-photon dissipation noise in 2-to-1 synchronization, without driving (E=0), with fixed parameters: $\Delta =0,\;\gamma_{1}=1,\;\gamma_{2}=10,\;p=0$. As discussed above, $MRL^{\left(1\right)}$ and phase coherence are both zero in these cases.

**Figure 7 entropy-25-01116-f007:**
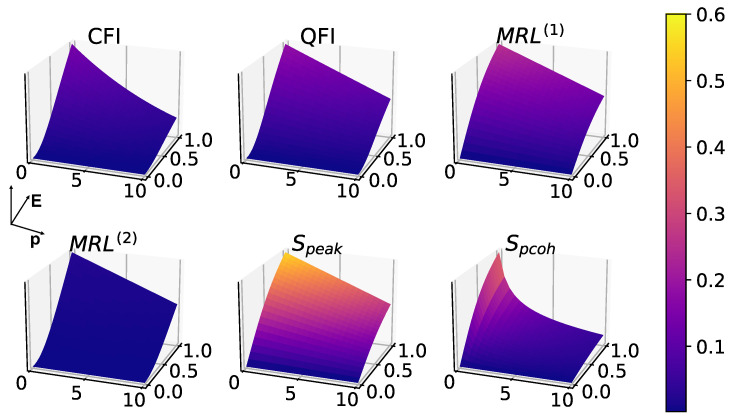
Effects of white noise in 1-to-1 synchronization, in the absence of squeezing (η=0), with fixed parameters: $\Delta =0,\;\gamma_{1}=1,\;\gamma_{2}=10,\;\gamma_{3}=0$.

**Figure 8 entropy-25-01116-f008:**
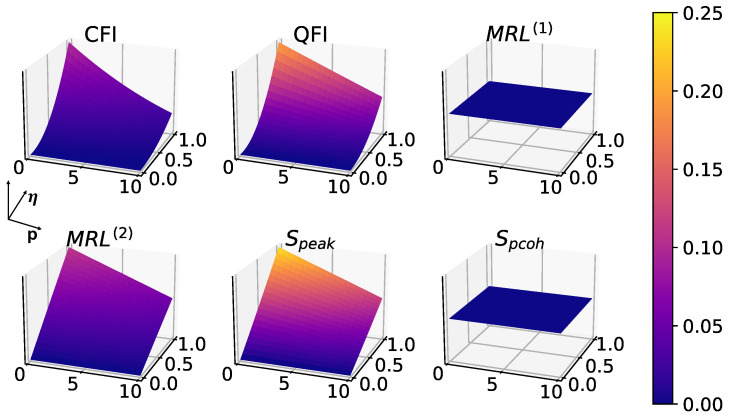
Effects of white noise in 2-to-1 synchronization, without driving (E=0), with fixed parameters: $\Delta =0,\;\gamma_{1}=1,\;\gamma_{2}=10,\;\gamma_{3}=0$. $MRL^{\left(1\right)}$ and phase coherence are 0 in this case for the same reason as above.

**Figure 9 entropy-25-01116-f009:**
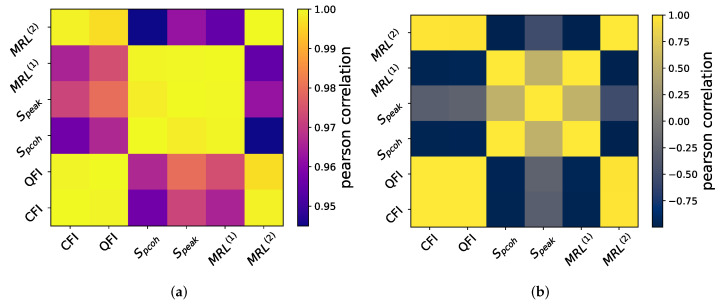
Correlation between different measures on (**a**) 1-to-1 synchronization. Calculations are performed on the same data as Figure 3—left column. (**b**) 2-to-1 synchronization. Calculations are performed on the same data as Figure 4—right column.

**Figure 10 entropy-25-01116-f010:**
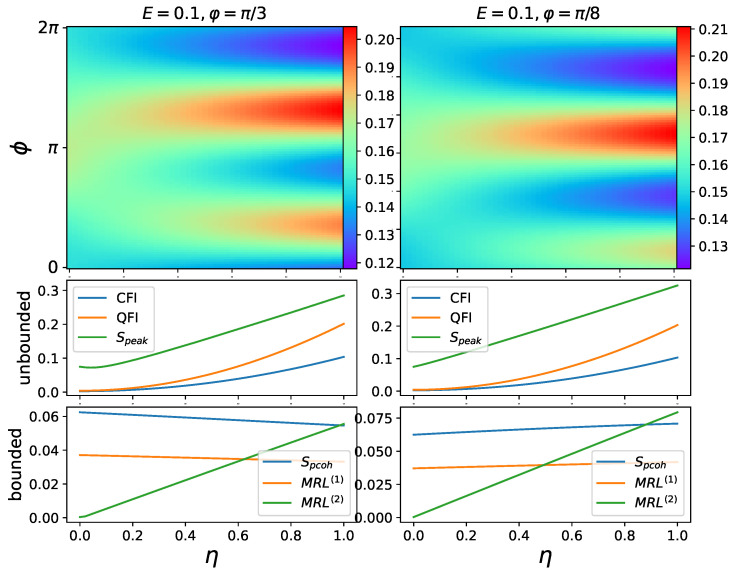
Asymmetrical phase distribution and synchronization measures plotted against squeezing η. Fixed parameters: $\Delta =0,\;\gamma_{1}=1,\;\gamma_{2}=10,\;\gamma_{3}=0$.

## Data Availability

Data available upon reasonable request.

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
