# Peer review of "Fisher Information as General Metrics of Quantum Synchronization"

_entropy, 2023, doi:10.3390/e25081116_

Round 1

Reviewer 1 Report

Dear Editor,
 it is for a pleasure to have read this manuscript sent by the authors to your paper.
The article seems well written and well structured to me.
I have a few comments:

1) I see that Fisher Information, classical and quantum, is very much present in a large number of publications. Perhaps the authors could more strongly emphasize the uniqueness of this work compared to the literature, or make it clear how this work brings forward a new idea in the use of Fisher Information.

2) Perhaps, the authors could add a sort of proof-of-principle picture that gives in a simple way the idea of the manuscript.

3) The authors could stress in more detail the classical and quantum Fisher information.

4) The authors could consider to enrich the literature.

Reviewer 2 Report

In the manuscript, the authors consider various measures of quantum synchronization for an externally excited oscillator and explore the possibility of introducing Fisher information to determine quantum synchronization, which is especially important for the case of a quantum problem. The general formulation of the problem seems to be quite relevant.

At the same time, while reading, significant questions arose:

1. The article does not contain a detailed description of the Stuart-Landau quantum oscillator model, the assumptions that were used in its formulation, and the physical motivation for considering this particular model by the authors is not given.

2. The equations of motion are written in such a compact form that it is not directly visible from it how the dynamics of the oscillator itself is described and how changes in the states of photons are taken into account, i.e. all degrees of freedom are presented at once in a collapsed form. In Appendix A, without sufficient explanation of the meaning of the individual elements of the density matrix and the method for obtaining the final formulas, only the limit solution is given. Thus, the possibility of reproducing the results of the work and using the apparently valuable methodology presented in it remains unclear.

3. The proposed formula (5) also needs a detailed discussion. Just a reference to the works that prompted the authors to such a choice of measure is clearly insufficient.

The noted shortcomings do not allow for the time being to give a reasonable conclusion about the advisability of publishing the article. The material presented needs to be substantially improved, since it is required to more fully disclose the content of the work itself and the methods, equations and procedures used in it.

Round 2

Reviewer 2 Report

In the manuscript, the authors consider various measures of quantum synchronization for an externally excited oscillator and explore the possibility of introducing Fisher information to determine quantum synchronization, which is especially important for the case of a quantum problem. The general formulation of the problem seems to be quite relevant.

At the same time, while reading, significant questions arose:

1. The article does not contain a detailed description of the Stuart-Landau quantum oscillator model, the assumptions that were used in its formulation, and the physical motivation for considering this particular model by the authors is not given.

2. The equations of motion are written in such a compact form that it is not directly visible from it how the dynamics of the oscillator itself is described and how changes in the states of photons are taken into account, i.e. all degrees of freedom are presented at once in a collapsed form. In Appendix A, without sufficient explanation of the meaning of the individual elements of the density matrix and the method for obtaining the final formulas, only the limit solution is given. Thus, the possibility of reproducing the results of the work and using the apparently valuable methodology presented in it remains unclear.

3. The proposed formula (5) also needs a detailed discussion. Just a reference to the works that prompted the authors to such a choice of measure is clearly insufficient.

The noted shortcomings do not allow for the time being to give a reasonable conclusion about the advisability of publishing the article. The material presented needs to be substantially improved, since it is required to more fully disclose the content of the work itself and the methods, equations and procedures used in it.